# Sustaining Equality and Equity. A Scoping Review of Interventions Directed towards Promoting Access to Leisure Time Physical Activity for Children and Youth

**DOI:** 10.3390/ijerph19031235

**Published:** 2022-01-22

**Authors:** Rikke Lambertz-Nilssen Hjort, Sine Agergaard

**Affiliations:** Department of Health Science and Technology, Aalborg University, 9000 Alborg, Denmark; sine@hst.aau.dk

**Keywords:** children, youth, leisure, physical activity, health equity, health equality, sport, social sustainability

## Abstract

Promoting access to leisure time physical activity (LTPA) opportunities for children and youth is crucial to promote public health. Yet, ensuring sustainable interventions in this field requires theoretically informed approaches to guide the processes of developing, implementing and evaluating LTPA programs. The objective of this review was to examine how concepts of equality and equity have been operationalized in LTPA interventions for children and youth in order to identify facilitating factors and barriers to LTPA access connected to such concepts. Using a pre-piloted search strategy, three electronic databases were searched for studies of interventions aiming to promote access to LTPA in organized or community sport for children and youth. Following a screening process, 27 publications representing 25 unique interventions were included. Through careful examination of the aim and target group of each intervention, they emerged in three categories in accordance with their (implicit) understandings of equality and equity. Also, considering the processes through which the interventions within each category had been implemented leads to the conclusion that an explicit theoretical understanding of the aim of interventions will increase communal knowledge among intervention stakeholders about which facilitating factors to pursue and barriers to bypass to contribute to more socially sustainable LTPA programs.

## 1. Introduction

It has been estimated that 80% of youth (11–17 years old) worldwide fail to reach the recommended minimum 60 min of moderate-to-vigorous intensity physical activity (MVPA) per day [1]. This is supported by the Global Matrix 3.0 Physical Activity Report Card Grades for Children and Youth on physical levels of children and youth from 49 countries reporting that only 27–33% of children and youth meet the physical activity recommendation of 60 min of MVPA per day [2]. Physical inactivity among school-aged children and youth has been found to be associated with adverse physical, mental, social, and cognitive health outcomes [3,4,5,6]. Therefore, promoting access to LTPA opportunities for children and youth as a preventative measure has become an important part of improving public health.

In addition to the general low levels of physical activity of children and youth, studies also show differences in child/youth physical activity (PA). Adolescents with higher socioeconomic status (SES) are more physically active than those with lower SES [7,8]. With regard to sports participation, Milanovic et. al. showed that 70.9% of children from families with low parental education spent less than 2 h/week on sports compared with 38.2% of children with high parental education [9]. Other important disparities in child and youth physical activity are connected to gender, ethnicity and physical or mental disability [10,11,12,13]. Reducing such inequities requires sustainable interventions that have “the ability to maintain programming and its benefits over time” [14].

While sustainability in general can be defined as something maintained over time, it can pertain to a plethora of arenas and is typically divided into three types: Environmental sustainability, economic sustainability, and social sustainability. While all three forms of sustainability are important in relation to leisure time physical activity (LTPA) programs, this review focuses on the social sustainability of interventions directed towards promoting equal access to physical activity and sports participation for children and youth. Vallance, Perkins, and Dixon defined social sustainability as a collective understanding of the need to build a community in which the participants thrive with equal access to opportunities for individual development [15]. Thus, social sustainability in LTPA interventions are strengthened by concepts and knowledge that can increase communal understanding (shared knowledge) of facilitating factors and barriers connected to various operationalizations of equity or equality in interventions directed towards promoting equal access for children and youth to participate in physical activity and sport.

This review examines the understandings inherent in interventions directed towards promoting access to LTPA for children and youth. A theoretical distinction between equity and equality will be used to explore the understandings and operationalization’s of such concepts along with the facilitating factors and barriers to equal access that appear when interventions set out to reach their respective aims. In so doing, the objective of this article is to contribute to the social sustainability of LTPA programs through strengthening shared knowledge (communal understanding) among intervention stakeholders about the consequences of implementing interventions according to principles of equity or equality. Such knowledge involves understanding which facilitating factors to pursue and barriers to reduce in order to promote access to LTPA for children and youth. To reach such a detailed understanding, this article is based on a systematic literature review of existing intervention studies directed towards promoting access to LTPA for children and youth.

### 1.1. Former Reviews

In the field of child and youth LTPA participation, former reviews have covered literature on conditions for child and youth LTPA participation as well as different target groups and intervention types.

One group of reviews have reported on conditions and interventions related to the promotion of LTPA to broad groups of children and/or youth. These reviews map barriers to children’s participation in sport [16] or explore reasons for their participation or non-participation [17]. Only one review has focused specifically on *interventions* to promote PA assessing the value of different components such as family/community involvement and activities before and after school in comprehensive school-based PA programs [18].

While these reviews help to illuminate conditions related to LTPA promotion to general target groups of children or children and youth, other reviews have focused on literature that explores efforts to include a specific target group in LTPA. These reviews are based on the understanding that specific target groups participate in LTPA to a lesser extent than other adolescents and need specialized interventions to provide equitable access to LTPA participation. Two reviews focus on efforts to include children with disabilities in out-of-school PA in general [19] or in sports clubs specifically [20]. Other reviews have focused on at-risk youth and assess the effect of sports-based youth development interventions [21,22].

No previous reviews have summarized the knowledge of how different theoretical understandings of equality and equity inform and affect interventions to promote child and youth LTPA participation. Further, no reviews have covered studies of such interventions with a variety of aims and target groups.

### 1.2. Equality and Equity

Theoretically, this review draws on perspectives from public health studies in which different concepts have been used to provide understanding of a variety of approaches that can shape interventions directed towards reducing unequal access to health [23,24,25]. When the attention is on health inequality, general differences in the health of individuals or groups are in focus, while an understanding of health *inequity* turns the focus to specific target groups with particularly poor health. Whereas health *inequality* may be observed without moral judgement, health inequity is considered unfair or unjust in the sense that it is preventable or unnecessary [26]. Especially our use of the concept of equity will benefit from further definition as there is an enormous amount of literature on equity in health and health care, and several principles of equity are commonly discussed [27,28]. At the most general level, equity in health care requires that patients who are alike in relevant respects be treated in like fashion (horizontal equity) and that patients who are unlike in relevant respects be treated in appropriately unlike fashion in proportion to the differences between them (vertical equity). One of the ways in which patients can differ is on the basis of need. In health and health care literature need is an ambiguous concept [29,30] but is most commonly taken as being proportionate to health itself [31]. In this view, the lower one’s health the greater one’s need, and combined with the concept of vertical equity those with greater need should receive greater attention and more resources. For the purpose of this study, the principle of distribution of resources according to need guides our definition of equity as it may also be observed in interventions that aim to promote LTPA for children and youth.

Since this study aims to explore the variety between the concepts of equity and equality, and how such understandings are operationalized in LTPA interventions, we are also inspired by respectively universal and targeted approaches to interventions which sit at the intersection of public health and social policy [32,33]. In public health, the population-level (universal) approach describes interventions delivered to whole populations while the high-risk (targeted) approach concerns individuals identified as having elevated risk of a particular health problem [34]. In our analyses interventions based on the general understanding of equality show the same characteristics as universal interventions as they aim to include a broad population such as “as many as possible” or “all children and youth” and offer every potential participant the same form of help or support to remove differences/inequalities in health. In comparison, interventions seeking to promote health equity are based on the understanding that differences in health stem from unjust societal structures causing discrimination or lack of access to certain resources for specific subgroups of the population. This perspective will often lead to a targeted approach directed towards the population segment(s) deemed to be vulnerable or disadvantaged as a result of varying societal resources such as economic status, education level, sex, place of residence, race, ethnicity, age, or disability status. Therefore, interventions based on an understanding of health equity will aim to address these underlying structures by providing additional resources that target the varied needs of disadvantaged subgroups.

## 2. Methods

We conducted a scoping review with the purpose of mapping the understandings that guide existing studies of interventions aimed at promoting access to LTPA for children and youth. The main characteristic of a scoping review, is that it provides an overview of a broad topic [35,36]. Arksey and O’Malley described scoping reviews as particularly suited for summarizing and disseminating a breadth of research findings and for identifying research gaps in the existing literature [37]. For our purpose, the scoping review provides a framework to describe the understandings that guide existing interventions and to group such findings into communicable categories. Furthermore, this review goes beyond descriptive reporting. It discusses the ways in which a more explicit use of theoretical understandings (e.g., of equality and equity) could contribute to a better understanding of the specific options (and barriers) that follow different approaches to interventions aiming to provide access to LTPA for children and youth. In this regard, whereas the literature search was primarily conducted by scoping review procedures, the subsequent discussion includes a narrative synthesis of the findings.

### 2.1. Search Strategy

By conducting a scoping review, we sought to gain insight into the current research (defined as research published over the years 2000–2021) on interventions to promote LTPA for children and youth aged 0–24 years. The search protocol was designed a priori and followed the PRISMA-ScR guidelines [38]. The first step was a creative search on Google, Google Scholar, and ProQuest using the terms child; youth; include; access; participation; equity; equality; sport; exercise; club; community; leisure. This initial search highlighted the need to broaden the search strategy and remove the facet equity/equality as it yielded very narrow results causing only studies with an explicit aim to explore or assess interventions from this perspective to be included. At the same time, studies on interventions aiming to promote “sport for all” that were not explicitly positioned in relation to inequality were omitted. This step also included analyzing words contained in the title and abstract as well as the keywords used to describe the article in order to identify relevant search terms.

In the second step, the facet equity; equality; inequity; inequality; difference was removed, and the search terms were used in a second search among the ScoPus, SPORTDiscus, and ProQuest databases. These databases were selected to cover literature on the research topic that we deemed to be within social sciences and sport sciences and because the databases (except from SportDiscus) contain a wide range of academic material/publications. The search was conducted in August 2020 with an updated follow-up search in August 2021. The search terms used in both of the searches are listed in Table 1.

After the removal of duplicates (*n* = 791), the number of articles included for screening was 5885. After title-abstract screening, 84 articles were included for full-text screening, and ultimately 27 were included in the final synthesis. Figure 1 shows the screening process.

All potentially relevant papers were read in full by the first author, and the selection was conducted in accordance with the inclusion and exclusion criteria described below. The process of including and excluding articles was discussed with the second author, who was also consulted regarding all papers that gave rise to any kind of uncertainty with the first author.

### 2.2. Study Selection and Data Extraction

The title and abstract of the identified papers were examined for relevance to the objective of the review. Studies were excluded if they did not meet the following inclusion criteria: (a) published in a peer-reviewed journal during or after the year 2000; (b) written in English or Danish language; (c) documented interventions or programs taking place in a western country; (d) the documented interventions or programs aimed to increase participation in organized physical activity in a sports club or other types of groups or communities (e) participants/target group of the intervention were children or youth under the age of 24 years. For this criterion, we adopted the United Nations’ definition of youth as being persons aged 15 to 24 years [39,40] meaning that it partly includes young adults (defined as those aged 20–29). Thus, Interventions targeting both youth and young adults were included in the review.

Thus, studies investigating various effects of LTPA participation and studies mapping the amount of PA among children and youth were excluded. Consequently, the review was directed towards studies that described the processes of developing, implementing, and evaluating interventions, thereby providing insight into the barriers and facilitating factors identified in former intervention studies. Former reviews were excluded due to the difficulty of gaining insight into the specific interventions that the studies described. Also, studies of health interventions with a main focus on increasing PA levels of children and youth in settings other than sports clubs or communities were excluded in order to maintain the focus on exploring various understandings of how to promote access to LTPA participation in a club or community.

Based on the theoretical understandings of health equality and equity as well as universal and targeted approaches to health care the 27 identified studies were categorized into three distinct types focusing on promoting: (1) equal access to LTPA, (2) equity through LTPA and (3) equity in LTPA. Using an inductive approach, we examined the intervention studies within each category and identified facilitating factors for children and youth to access LTPA along with barriers to access within each study. These facilitating factors and barriers were then synthesized thematically across the studies in each category.

Table 2 shows the objectives of the selected studies, as well as characteristics, target group and aim of the interventions assessed in each study.

## 3. Results

As previously described when presenting the theoretical framework, the identified interventions turned out to be distinct based on the following: (1) they promoted general access to LTPA for broad target groups of children and youth in accordance with understandings of equality or (2) they aimed to reduce unequal access for specific target groups in accordance with understandings of equity. A further division between equity through and equity in LTPA was made to distinguish between interventions with similar target groups but different aims. Thus, ‘equity through LTPA refers to interventions that aim to use LTPA participation as a tool to promote the health, wellbeing, and social inclusion of specific target groups. ‘Equity in LTPA’ interventions aim to promote access into LTPA for specific target groups that are particularly underrepresented there. In the sections below, we will present the different understandings of equity/equality that frame each category along with the facilitating factors and barriers connected to this understanding.

### 3.1. Equal Access to LTPA

Ten studies covering nine unique interventions were allocated to this category (Table 2). Interventions in this category aimed to promote access to LTPA by making resources and activity options available to broad, general target groups defined as young people [41], youth [42], school-aged children or children at a specific age [43,44], or children in specific grades or levels at school [45,46,47,48]. The aims of the interventions described in the studies were to engage unassociated members of target groups in organized sport or to increase PA (in a club or community setting). Only one study specifically stated that the intervention aimed to promote participation for unassociated youth [49], otherwise interventions did not differentiate between children or youth who were already engaged in LTPA and children/youth who were not.

Since the aim of the interventions (as described by the studies included in this category) was to increase general LTPA participation, consequently the focus was on developing and implementing activities to increase the volume of LTPA opportunities available to the broad target group. Thus, promoting access to LTPA for children and youth in accordance with understandings of equality turns the attention to increasing opportunities available to a large target group, irrespective of differences in resources, needs, and LTPA participation/experience. Such processes are connected with both facilitating factors and barriers in promoting access to LTPA for children and youth.

#### 3.1.1. Facilitating Factors

##### Volume/Availability

Of the studies allocated to the equality category, six studies reported on interventions that were organized as national programs or policies [41,42,44,47,48,49] and four studies reported on regional/statewide interventions [43,45,46,50]. All of these interventions were organized around multiple local partnerships in order to provide as many activity options as possible. In other words, interventions with an equality perspective facilitated LTPA participation through volume defined as multiple pathways into LTPA participation. As well as focusing on recruiting a large number of children and youth, the interventions described within the studies belonging to this category were also directed towards recruiting a large number of local partners or access points in order to increase availability (seven studies) [42,44,45,46,47,48,49].

In accordance with the equality understanding, the studies grouped into this category described a wide range of aims to promote access for broad target groups containing a wide variety of children and youth. Consequently, interventions within this category were designed to give everyone in the target group the possibility to participate regardless of their socio-economic resources and mobility. Four studies reported on interventions that situated intervention activities at schools, thereby providing better opportunities for all children and youth to participate regardless of whether they had the necessary resources to transport themselves to LTPA in club and community settings or not [43,44,47,48]. Three studies reported on two interventions that offered activities with no participation fee making LTPA available for children and youth regardless of their socio-economic resources [45,46,50]. One of these studies showed that providing a free access pass to everyone in the target group did not mean that children living in neighborhoods with low socioeconomic status used the access pass more than target subgroups from other neighborhoods; participation rates were still higher in affluent neighborhoods [46].

Such results also indicate that while it might be considered a strength to include large target groups and make resources available to as many as possible, one of the weaknesses might be that the effect of the intervention was not as initially intended.

#### 3.1.2. Barriers

##### Complexity of Partnerships and Communication

The need to form several local partnerships to generate volume in these interventions adds to the complexity of implementing the intervention. Five out of 10 studies reported difficulties with intervention implementation when the partners held different perspectives on the aim and the values which should guide intervention partnerships [41,43,44,47,49]. In general, it was difficult to communicate the aim and values of the program to many partners and to ensure that these were practiced accordingly in the intervention. In addition, studies pointed to differences in perspectives and resources between the professional teachers and volunteer coaches involved as well as to differences in interests between national sports organizations, local schools, local sports clubs, and national policy makers.

##### Recruiting over Change

There was a strong focus on the process of recruiting participants and providing access through transportation, economic resources, information, instruction, or equipment in interventions in this category. However, none of the studies in this review reported on interventions for broad target groups aiming to make qualitative changes in existing sport activities. Rather, the focus was on how activities were organized and promoted, more than on developing the actual activities. One study pointed specifically to the strong focus on recruiting as a barrier in this type of intervention, arguing that volume was preferred over qualitative changes [49]. The efforts were concentrated on providing access as opportunity by enhancing availability through activity passes or school partnerships that made it easier across target groups to physically get to the activity. This focus can lead to new ways of recruiting and promoting LTPA interventions, but it does not necessarily lead to changes or even an increase in the number of children or youth utilizing access if the barrier to participation should be found in the activities in which the participants are expected to partake.

##### Lack of Knowledge about Target Groups and Involvement

Interventions in this category do not apply much prior knowledge on the needs, interests, and resources of specific target groups. Of the 10 studies in this category, none involved the target groups’ perspectives and did not provide them with options of giving feedback and contributing to the design of the intervention. Instead, most (seven) of the studies focused on the partnerships between the organizations involved in the delivery of the intervention [41,42,43,44,47,48,49].

##### Reproducing Social Inequalities

Two studies of interventions that provided free access passes/vouchers to LTPA indicated that children and youth from the most disadvantaged segments of the population/low-income neighborhoods were less likely to redeem the voucher or make use of the access pass than those living in neighborhoods with higher socioeconomic status [46,50]. Consequently, a possible barrier in interventions with general target groups was that social inequalities might be reproduced rather than challenged.

### 3.2. Equity through LTPA

Eleven studies covering 10 unique interventions fell into this category (Table 2). These interventions targeted specific groups who were perceived to be unjustly positioned to gain access to resources in society. The aim of these interventions was to reduce unequal access to not only LTPA but to general societal resources by using LTPA participation as a tool to promote health, wellbeing, and social inclusion for specific target groups.

The specific target groups in the studies were young people not engaged in sport [51], unassociated youth in deprived residential areas [52], migrant and refugee children and youth [53,54], minority youth [55], migrant youth in socially disadvantaged areas [56], underprivileged youth [57,58], children and youth in socially disadvantaged areas [59,60], and young people experiencing psychological and/or social problems [61]. Five out of 10 interventions aimed to engage the target groups in LTPA activities as an alternative to criminal or “rowdy behavior” during weekends or holidays [52,55,56,57,58,59].

The element of close proximity was key in nine out of 10 interventions in this category in order to establish a connection to local youth perceived as needing help to establish healthy relationships to both adults and other youth [52,54,55,56,57,58,59,60,61].

#### 3.2.1. Facilitating Factors

##### Utilizing Local Knowledge and Resources

Four out of 10 interventions in this category had a specific focus on local accessibility in the sense that they were situated in spaces inhabited by the target group, such as schools, community sites, or spaces around and in between housing blocks [54,56,59,60]. If interventions are developed locally they can benefit from local knowledge about children and youth living in the specific context-where they meet, what they do, and how–or who–to best approach/recruit them. Six out of 11 [54,55,58,59,60,61] studies described how community leaders or outreach workers were used in developing connections to the target groups at the intervention sites. These elements of having close proximity to the target groups and utilizing adults with a local connection represented two strategies to increase accessibility to activities for the specific target group.

##### Local Ownership and Involvement

Equity through sport interventions are local in the sense that they are designed to meet a local need for alternate activities and alternate arenas of LTPA where local youth can become included into the communities. Four out of 10 interventions directly arose from local initiatives to engage youth in LTPA [59,60,61], while six out of 10 were a part of a regional or national initiative [51,52,53,54,56,57,58]. However, all 10 interventions were embedded locally, and target groups were defined by their connection to a specific neighborhood/housing area/local community. All interventions had local partners involved in activities, such as local clubs, local leaders, volunteers, and outreach workers. As described above, in addition to strengthening the intervention ability to reach the specific target groups and meet their needs, this also shows a potential for these interventions to inspire local ownership and empowerment in the process of making a change for local youth. Rather than having local actors with limited influence adapting national interventions local stakeholders and partners in these locally developed interventions have ownership of the resources going into the intervention, as well as knowledge generated through the interventions.

#### 3.2.2. Barriers

##### Unintentional and Undetected Exclusion of Subgroups

Three out of 11 studies showed that youth who had no previous knowledge of or experience with participating in sport clubs and specific LTPA activities were likely to be or feel excluded from such activities [51,52,53]. This was due to their lack of knowledge of the expectations inherent in participating in LTPA and club sports, such as which equipment to use or proper etiquette for a given sport. Also, the target group often held previous experiences of failing or not being good at sports [52]. In interventions open to all children or youth in a specific housing area, these exclusionary mechanisms may go undetected by volunteer coaches (who themselves have access to knowledge and positive experiences with LTPA participation) and government officials (who are too far removed from the intervention to observe the exclusionary mechanisms at play).

One study questioned the assumption that sports clubs are ideal arenas for the integration of migrant youth [53]. This study found that the discourses constructed in the intervention in focus created very little sense of belonging among refugee youth, at times rather a sense of alterity, and created few social spaces for ‘bridging’ between majority and minority groups. The language of ‘us’ and ‘them’ was unintentionally and unconsciously maintained within the sports club setting, leading to a reproduction of social inequalities rather than these being evened out through sports club participation.

##### Sport as a Tool Rather Than Sport for Its Own Sake

Three out of 11 studies showed how interventions in this category used LTPA as a means for societal objectives rather than an end in itself to promote feelings of autonomy and community. As such, LTPA participation was phrased as a tool for personal development and social inclusion and as a safe space for youth living in disadvantaged areas, and this influenced the way the participating children and youth experienced LTPA. In addition, the personal connections created through the intervention were instrumentalized as tools to develop engaged and law-abiding citizens and personal growth [56,57,58]. While participants might have expressed joy in playing with teammates and improving their skills, they were also aware that one of the main factors driving the interventions was the need to be safe; this was expressed in interviews with the target group in two of the studies [55,57].

### 3.3. Equity in LTPA

Six studies on six unique interventions were allocated to this category (Table 2). These interventions aimed to reduce unequal access to LTPA for a specific target group that was underrepresented in LTPA. In this review, we have identified studies of interventions in which the target groups were girls [62] and women [63], children and young adults with disabilities [64], ethnic minority youth, disabled young people, youth from socioeconomically deprived backgrounds [65], children in grades 3–5 in economically vulnerable neighborhoods [66] and looked after children [67].

The interventions that fell into this category distributed intervention resources according to the perceived needs of the target group(s). This rested on the premise that there were structures in the society and specific sports/activities that led to unfair differences in accessibility and participation in LTPA for specific target group(s). While interventions in the previous category aimed to achieve equity for the target group *through* LTPA participation, interventions in this category strove for equitable participation *in* LTPA.

#### 3.3.1. Facilitating Factors

##### Targeting Resources towards Social Justice

All six interventions in this category focused their resources on a specific target group and on reducing the barriers to access that arose in their situation. Two interventions targeting girls strategically used young women as role models in the design to address the low visibility of girls in the activities [62,63]. An intervention targeting children and youth with disabilities used able-bodied buddies to support the target group [64]. All six interventions grouped in the equity in sport category took the needs of the specific target groups as their starting point and allocated resources to the specific individuals considered to be most in need. As such, these interventions appeared to be directed towards contributing to greater social justice in LTPA.

##### Empowerment/Ownership

Two out of six interventions in this category were based on target group involvement and ownership; both were interventions to reduce unequal access for girls/young women [62,63]. The remaining interventions in this category were led by teachers [65,66] or developed and implemented by professionals with a prior knowledge of the specific needs of the target group [64] or by government officials [67]. Of the two interventions that involved the target group, one did so by asking the participants to express their needs and preferences in the process of designing the intervention [62], and the other was initiated and led by target group representatives (women skateboarders) [63]. Thus, various efforts were made to promote empowerment and ownership within the target group or in close proximity to them.

#### 3.3.2. Barriers

##### Narrow Scope

Interventions targeting specific groups can be viewed as too narrow in scope, allocating too many resources to a relatively small target group. This can be perceived as unfair, especially if it is implemented in partnership with institutions such as schools where the attention of teachers is usually directed towards including all children [65].

##### Stigma

Interventions in this category can potentially increase the stigmatization of the target group by making them highly visible as a group in need of extra resources to reduce unequal access. This was specifically addressed in two studies of interventions that provided girls only training sessions [63,65]. Girls would then be constructed as inferior in skills and resources compared to boys that were not in need of such special resources [63]. Another study pertained to an intervention that targeted children with disabilities by providing mixed training sessions with special support for the target group. This enabled access to mixed sessions, but also made the target group highly visible [64].

## 4. Discussion

None of the studies identified through our scoping review reported on interventions presenting an explicit understanding on equality or equity. Conversely, it is clear across the interventions presented in our review that there were many different understandings of the aim of the interventions, especially in interventions with several partnerships involved. However, theoretical perspectives on equity or equality can provide a common lens–a communal understanding–through which relevant facilitating factors and barriers can be identified and acted on for the partners working together on these interventions.

For interventions promoting ‘equality in LTPA’ to become more sustainable, the partners involved should not only focus on the facilitating factors of volume and availability but also work to avoid the barriers of communication in the complex partnerships needed to increase the volume. In addition, they should avoid tendencies towards focusing on recruitment and rather focus on changing sports activities and make receiving settings more inclusive.

In interventions promoting ‘equity through LTPA’, the facilitating factors identified are the close proximity of the intervention to the specific target group and its embeddedness in a specific community or neighborhood. For such interventions to become more socially sustainable, attention should be on making the expectations inherent in participating in LTPA and club sports explicit among coaches and volunteers to enable them to support participants with no prior experience or bad experiences from LTPA participation. Furthermore, a communal understanding among intervention partners of the consequences of using sport as a tool rather than an end in itself should be established.

Finally, it is relevant to interventions promoting ‘equity in LTPA’ to involve the specific target groups, local resource persons, and institutions to promote a communal understanding and thereby contribute to the social sustainability of the interventions. As identified when reviewing the second category of studies, awareness of the weaknesses of the equity approach may also contribute to avoiding unintended stigmatization and exclusion of individuals not familiar with the involvement processes and activities facilitated by or with sports club volunteers/coaches.

While equality and equity are theoretical definitions, when operationalized into these three categories, they can help to guide interventions and contribute to their social sustainability even if the challenges to social sustainability are many. Such theoretical knowledge enables project managers to design and adjust interventions to pursue relevant facilitating factors or avoid related barriers to LTPA programs. Another theoretical perspective that might be useful to explore in such interventions is involvement of stakeholders and the target group of children and youth as it is a theme that emerges as important within all three categories even if it is sometimes present and other times absent.

For the purpose of developing a conceptualization of equity and equality in LTPA interventions based on empirical evidence, the categories and themes were brought forth through an inductive analysis of the studies. While this approach connects the concepts of equity and equality to practice in LTPA interventions, it also makes the analysis dependent on the perspectives/content of the 27 studies included in the review. The insights from this review could be further developed and qualified by connecting them with other established theoretical frameworks. One such could be the Social Determinants of Health framework [68,69] which is a framework that could further address various dimensions of equity (as income, transportation, time, etc.) in planning interventions aimed at increasing access to and reducing barriers for LTPA.

The studies included in the literature review presented in this article were screened with the specific aim of gaining insight into how understandings of equity and equality were reflected in the interventions. Consequently, studies focusing on possible effects of the interventions on the PA levels of children and youth were excluded in the review. This means that we cannot compare how successful the different types of interventions were. Rather, our focus is on the processes through which the interventions were developed and implemented, specifically on exploring the facilitating factors and barriers in promoting access to LTPA for children and youth in detail.

In future studies, it is relevant to gain further insight into such facilitating factors and barriers when they arise during the complex processes of developing, implementing, and evaluating interventions. This observation encourages us to shift towards using methods such as participatory action research [70,71]. This method allows researchers to work with project managers to not only operationalize understandings of equality and equity but also to develop practices to pursue the facilitating factors and confront barriers as they arise in interventions.

## 5. Conclusions

This review explored how different theoretical understandings of equality or equity were operationalized in existing intervention studies on how to promote access of children and youth into LTPA. The results show that the approaches in the studies identified through our scoping review fell into three main categories (1) equality in LTPA, (2) equity through LTPA, and (3) equity in LTPA. When grouping existing interventions into these categories, we identified a range of facilitating factors as well as barriers to promoting access to LTPA for children and youth.

Our review also shows that very few interventions worked from an explicit understanding of equality and equity and thereby without awareness of the facilitating factors and barriers related to the specific understanding. To create and increase the social sustainability of interventions that take diverse approaches to promoting access to LTPA for children and youth, we recommend that the partners involved in each intervention develop a communal understanding of how they will profit from the facilitating factors and bypass the barriers associated with their specific approach.

## Figures and Tables

**Figure 1 ijerph-19-01235-f001:**
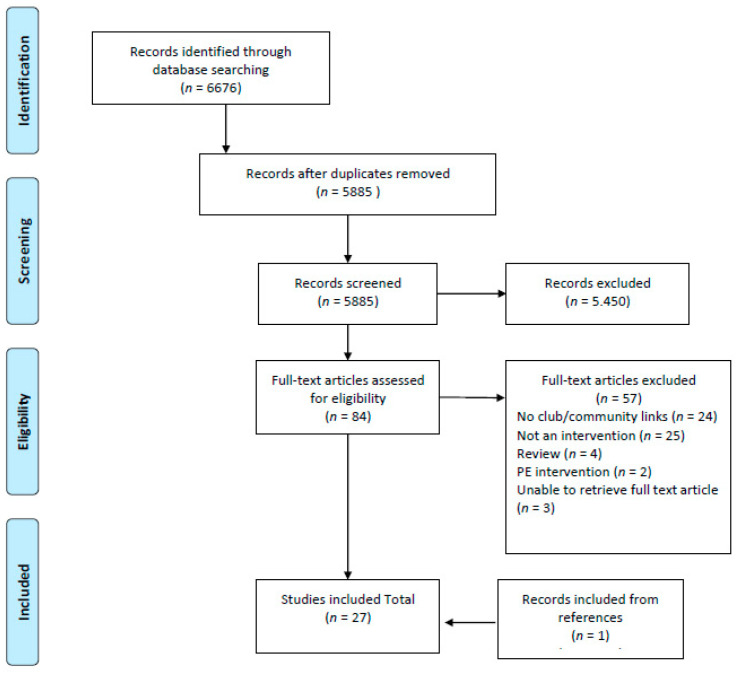
Flowchart of the screening process.

**Table 1 ijerph-19-01235-t001:** Search terms.

	AND			
OR	(a) Facet	(b) Facet	(c) Facet	(d) Facet
	Access Participation Inclusion Exclusion Include Involve	Sport “Physical activity” Exercise “Physical exercise”	“Sport association” “Sport organization” “Sport organisation” Club Society Community Leisure Recreation	Child Youth Adolescent Boy Girl Teen Young adult Kid

**Table 2 ijerph-19-01235-t002:** Data extraction chart: Summary of Study objective, intervention characteristics, target group and aim.

Authors	Country	Study Objective	Intervention	Target Group	Intervention Aim
EQUALITY					
Equal acces to LTPA					
Clark et. al., 2018	Canada	To evaluate the uptake of ACT-i-Pass, and understand the extent to which the intervention provides equitable access to children.	ACT-i-Pass provides free access to PA opportunities (facilities/programs).	Grade 5 students in London (Ontario).	Increase PA levels among children.
Clark et. al., 2019	Canada	To examine factors influencing the use of a free community wide physical activity access pass.	ACT-i-Pass provides free access to PA opportunities (facilities/programs).	Grade 5 students in London (Ontario).	Increase PA levels among children.
Eime & Payne 2009	Australia	To explore the structural links between participation programs conducted in schools and participation in community-based sporting clubs.	State Sports Governing organizations funded by a government funded health promotion organization to develop and deliver school-and community-based programs.	School-aged children and youth in Victoria.	Promote community-level sports participation.
Karp, Fahlén & Löfgren 2014	Sweden	To discuss mechanisms of change and inertia in Swedish sports by applying path dependency theory on results achieved in Idrottslyftet.	Sports clubs apply for funding for club activities and projects from national program (idrottslyftet).	Children and youth (especially from underrepresented groups).	Engage more children and youth in organized sport.
Keat & Michael 2013	New Zealand	To analyze the impact of changes in national sport policy on regional sports trusts.	National initiative (Kiwisport): Regional sports trusts asses community needs and provide funding to community organizations accordingly.	School-aged children	Get more school-aged children involved in organized sport.
Parnell et. al., 2015	England	To explore the delivery and partnerships involved within the School Sports Premium.	National strategy: Physical Education and School Sport (PESS). PE and school sport delivered by professional football clubs.	Young people	Promotion of PA and lifelong participation.
Ramanathan et. al., 2018	Canada	To evaluate the ParticipACTION Teen Challenge micro-grant program.	National micro-grant scheme (participACTION): Small budgets of money awarded via grant applications to community organizations.	Youth	Increase youth PA
Reilly et. al., 2021	Australia	To describe the uptake of Active Kids and assess the impact of the scheme on organized sport participation and child physical activity in a region of New South Wales.	The AK scheme is a four year investment of more than $200 million, to help families across the entire state meet the cost of getting children into organized sport and recreation activities.	Children aged 4,5–18 years of age.	Help families meet the cost of getting children into organized sport.
Stylianou, Hogan & Enright 2019	Australia	To examine the enactment of Sporting Schools program from the perspectives of sporting organizations, coaches and teachers.	National program: national sports organizations deliver school sports programs through community sports clubs, coaches and private providers.	Primary school students (children)	Increase children’s sport participation and connecting children with community sport.
Tomik 2008	Poland	To characterize the activities of SSCs from the perspective of representatives of Polish Sports association.	School sports clubs organize sports activities and events at school in student’s leisure time.	All students	Increase LTPA for students.
EQUITY					
Equity through LTPA					
Agergaard, Michelsen & Gregersen 2016	Denmark	To contribute to an understanding of the rationalities of specific political interventions, and the techniques used to monitor the leisure activities of particular target groups using a governmentality perspective.	Non-for-profit organizations providing drop-in sporting activities in holiday periods.	Migrant youth in socially disadvantaged areas.	Offer organized leisure activities to children and adolescents in specified areas as a means of crime reduction and anti-radicalization.
Dángelo, Corvino & Gozzoli	Italy	A case study that explores the impact of a multi-stakeholder sport initiative developing social inclusion for socially vulnerable youth and the related challenges.	The sport-based program providing weekly soccer training sessions.	Young people experiencing various psychological and/or social problems.	Promote social inclusion through sport.
Dowling 2020	Norway	A micro-analysis of the ‘slippage’ between government visions of sport for integration for refugees and the local, contextual interpretations of sport policy for inclusion.	Voluntary sports club provide two weekly activities: drop-in football and a fitness-and -motor skills training.	Unacompanied youth refugees.	Enabling youngsters to be integrated into the sports club while simultaneously lowering the threshold for making contact between refugees and locals.
Ekholm & Dahlstedt 2018	Sweden	Case study. Examines, from a governmentality perspective, how supportive community actors conceptualize their charitable contributions, enabling opportunities for under-privileged youth to participate in sports.	Sports-based interventions run in partnership between a national foundation, local sports clubs and an elite football club. Activities are organized-yet spontaneous–five-a-side football on Saturday nights.	Under privileged youth in suburban residential areas of exclusion.	Promote integration through sport’ and ‘to develop a sense of responsibility and participation in society as well as employability […], to prevent social exclusion [and] to contribute to crime reduction.
Ekholm & Dahlstedt 2021	Sweden	A single-case study, examines how socio-pedagogical rationalities and technologies are articulated in discourse and assumed to operate within the intervention, and how certain ideals of conduct and social inclusion are represented in discourse.	Sports-based interventions run in partnership between a national foundation, local sports clubs and an elite football club. Activities are organized-yet spontaneous–five-a-side football on Saturday nights.	Under privileged youth in suburban residential areas of exclusion.	Promote integration through sport’ and ‘to develop a sense of responsibility and participation in society as well as employability […], to prevent social exclusion [and] to contribute to crime reduction’.
Fahlén 2017	Sweden	Show how the corporal character of activities commonly provided in sports-based policy interventions has implications for the results of policy implementation.	National program: Clubs organize spontaneous sports in ’drop-in’ sessions, focus on non-competitive sports and participants’ wishes.	Unassociated youth in deprived residential areas.	Usher unassociated youth into participation in regular sport club activities–away from rowdy behavior during weekends.
Jacobs, Castañeda & Castañeda 2016	USA	Not stated in paper.	Open gym basketball activities on Saturdays and open tournament activities on weekends and summer holidays.	Individuals between kindergarten and post-college in socially disadvantaged neighborhoods in Chicago.	Developing youth to enjoy sport and thrive in their homes, schools and community.
King & Church 2015	UK	Explore experiences of youth mountain bikers to provide insights into complexities of adopting lifestyle sports as a tool for inclusive practices in delivery of policy for sport and health.	Government initiative: Provision of facilities for mountain biking, open to all.	Young people under 16 (among other socially disadvantaged groups).	Increase social inclusion of young people through access to, and participation in, mountain biking activities.
Parent & Harvey 2017	Canada	Asses the partnership component of a community-based youth sport for development program, in order to contribute to knowledge about the conditions needed for positive outcomes in such programs.	Community-based youth sport for development program providing recreational and educational activities accessible to all in schools, community organizations and centers.	Children aged 6–12 years in an underprivileged community in Ottawa.	Social development through sport for youth (aged 6–12).
Rosso & McGrath 2016	Australia	Report on a pilot project of a sport-based community program.	Sports-based intervention under regional program (Football United). Regular, free soccer activities at school and community-based sites.	Migrant and refugee children and youth.	Promote health, wellbeing and social inclusion through football.
Stodolska et. al., 2014	USA	Explore factors that affect minority youths participation in an organized sports program from a socioecological perspective.	Three divisions for baseball and two for softball targeting minority youth.	Minority youth	Lead youth away from street life via participation in baseball and softball.
Equity in LTPA					
Bäckström & Nairn 2018	Sweden	Explore paradoxical spaces of two strategies to increase girl/women’s participation in Swedish skateboarding.	National intervention applying two strategies to create gender equity: (1) Strategic visibility through girls only events and training sessions, and (2) strategic entitlement through books, articles and a documentary portraying girls as athletes equal to/the same as, boys.	Girls and women	Create equitable access to skateboarding for girls and women.
Cunningham & Warner 2019	USA	Examine the factors that influence participation in a community program.	A unified community sports program, representing a sports model focused on serving all. Buddies (able bodied youth) support and follow the players throughout the activities.	Children and young adults with disabilities.	Enhance leisure participation among children and young adults with disabilities, alongside their able-bodied peers.
Flintoff 2008	UK	Explore the ways in which gender equity issues have been addressed in official texts, how they have shifted over time and how teachers respond to them in daily practice.	School Sport partnership–A number of schools working together to develop networks and opportunities between school PE and wider community and leisure and sport contexts.	Girls/young women, ethnic minority youth, disabled young people and youth from socio-economically deprived backgrounds.	Increase activity levels of previously underrepresented groups and make links between PE and out of school sports participation.
McNeil et. al., 2009	Canada	Identify if outreach support increases school-aged children’s participation in recreational activities.	Children in intervention schools were assigned a connector (outreach worker) to facilitate participation in recreational activities.	Children in grades 3–5 in economically vulnerable neighborhoods.	Increase school aged children’s participation in recreational activities in economically vulnerable neighborhood.
Morgan et. al., 2019	UK	Gather views from girls, teachers, stakeholders and parents to co-produce a multi-component school-based, community linked PA intervention.	Use of role modelling and increase awareness of opportunities for community PA.	Girls	Increase PA levels and promote sustainable changes in PA participation among preadolescent girls
Murray 2013	UK	Study on looked after children’s involvement in PA and sport by analyzing data from Freedom of Information requests.	Government expectation that local authorities offer free leisure provision in the form of access passes or by subsidizing leisure activity including sport.	Looked after children	Provide access to leisure activities for looked after children, equal to their peers.

## Data Availability

The search documentation that supports the findings of this study is available from the corresponding author upon request.

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
