# Peer review of "Sustaining Equality and Equity. A Scoping Review of Interventions Directed towards Promoting Access to Leisure Time Physical Activity for Children and Youth"

_ijerph, 2022, doi:10.3390/ijerph19031235_

Round 1

Reviewer 1 Report

Based on the social justice concepts of equality and equity, the present paper reviews interventions directed at promoting access to leisure time physical activity (LTPA) and categorizes them into three categories: (1) Interventions directed at promoting general access to LTPA for everyone (equality), (2) interventions for groups that are deprived of LTPA (equity through LTPA), (3) interventions directed at reducing unequal access to LTPA in disadvantaged groups (equity in LTPA). Potentials and barriers of the interventions were extracted from the reviewed articles and translated into recommendations how to improve interventions and their promotion.

The review in general and the identified strengths and limitations of the collected interventions in particular are certainly valuable towards improving the design of future LTPA programs as well as the targeted recruitment of participants. Moreover, the categorization of programs into the three categories was helpful for the identification of potentials and barriers.

My major concern with the paper is a conceptual concern. Equality and equity are social justice concepts that were introduced decades ago into the social justice literature and have been used fairly consistently since. The concept of equity was introduced by John Stacy Adams (1963, 1965). His equity theory claims that outcomes (such as pay for work) are perceived as fair if they are correspond proportionally to recipients’ contributions (such as achievements). Violations of the equity principle necessarily imply unfair advantages (relative overpay) for some recipients and unfair disadvantages (relative underpay) for others. Morton Deutsch (1975) argued that equity may not the only principle of distributive justice. In some exchange contexts and relationships, people consider outcomes as fair even if they deviate from the equity principle. Specifically, inequitable distributions can be perceived as fair if they are consistent with the equality principle (every recipient gets the same) or the need principle (outcomes are proportional to needs). Adam’s definition of equity and Deutsch’s definition of equality and need have been widely accepted in social justice theory and research (cf. Sabbagh & Schmitt, 2016). Importantly in the context of this review, it seems to me that the authors confuses equity with need. This conceptual inconsistency with the predominant definition of equity and need in the social justice literature should either be avoided or justified convincingly.

Although less crucial than the issue of conceptual consistency, I wonder about the advantage of categorizing intervention programs according to concepts of distributive justice. Why is this categorization most insightful? Why is it more insightful than other possible categorizations would be such as the activity profile of programs or the similarity of activities to popular sports or the diversity vs. uniformity of activities or group activities vs. individual activities? For example, the similarity of the activity profile to popular sport may have stronger effects on the social status of a program and its sustainability than the similarity of the activity profile to a less known and less popular sport.

References

Adams, J.S. (1963). Toward and understanding of inequity. Journal of Abnormal and Social Psychology, 67, 422-436.

Adams, J.S. (1965). Inequity in social exchange. In L. Berkowitz (Ed.), Advances in Experimental Social Psychology (Vol. 2, pp. 267-299). New York: Academic Press.

Deutsch, M. (1975). Equity, equality, and need: What determines which value will be used as the basis of distributive justice? Journal of Social Issues, 31, 137-149.

Sabbagh, C. & Schmitt, M. (Eds.) (2016). Handbook of Social Justice Theory and Research. New York: Springer.

Author Response

Dear reviewer,

Thank you for giving us the opportunity to improve our manuscript. We very much appreciate the time and effort that you have dedicated to providing valuable feedback on the manuscript. We have highlighted the  changes that your feedback have led to within the manuscript. Here is a point-by-point response to your comments and concerns.

Comments from reviewer 1

Comment 1: My major concern with the paper is a conceptual concern. Equality and equity are social justice concepts that were introduced decades ago into the social justice literature and have been used fairly consistently since. The concept of equity was introduced by John Stacy Adams (1963, 1965). His equity theory claims that outcomes (such as pay for work) are perceived as fair if they correspond proportionally to recipients’ contributions (such as achievements). Violations of the equity principle necessarily imply unfair advantages (relative overpay) for some recipients and unfair disadvantages (relative underpay) for others. Morton Deutsch (1975) argued that equity may not be the only principle of distributive justice. In some exchange contexts and relationships, people consider outcomes as fair even if they deviate from the equity principle. Specifically, inequitable distributions can be perceived as fair if they are consistent with the equality principle (every recipient gets the same) or the need principle (outcomes are proportional to needs). Adam’s definition of equity and Deutsch’s definition of equality and need have been widely accepted in social justice theory and research (cf. Sabbagh & Schmitt, 2016). Importantly in the context of this review, it seems to me that the authors confuse equity with need. This conceptual inconsistency with the predominant definition of equity and need in the social justice literature should either be avoided or justified convincingly.

Response: Thank you for raising an important point here. We have revised the section defining equity and equality in order to emphasize that equity and need are not interchangeable concepts, (section 1.2, page 3, lines 112-125). However, as our study is concerned with health promotion we follow the general definition of equity in health care (patients who are alike in relevant respects be treated in like fashion and patients who are unlike in relevant respects be treated in appropriately unlike fashion in proportion to the differences between them). In this context need is one of the relevant factors used to determine receipt of health care. As our aim is to distinguish between interventions with a universal approach (equality), and interventions with a targeted approach (equity), rather than exploring different definitions or understandings of equity, we focus on the principle of distribution according to need as a lens through which we observe equity, with the awareness that there are other lenses (principles) which could also be applied.

Comment 2: Although less crucial than the issue of conceptual consistency, I wonder about the advantage of categorizing intervention programs according to concepts of distributive justice. Why is this categorization most insightful? Why is it more insightful than other possible categorizations would be such as the activity profile of programs or the similarity of activities to popular sports or the diversity vs. uniformity of activities or group activities vs. individual activities? For example, the similarity of the activity profile to popular sport may have stronger effects on the social status of a program and its sustainability than the similarity of the activity profile to a less known and less popular sport.

Response: This is also an important reflection. As equity in health is established as an effort of international importance, and states and national and international organizations are increasingly implementing health equity policies, the decision to categorize and explore interventions according to the concepts of equity and equality is based in our aim to contribute to understanding the variety of approaches to health promotion and LTPA. Consequently, while other categorizations such as the diversity vs. uniformity of activities or group activities vs. individual activities might indeed be insightful in order to assess intervention effects, it would not make the same contribution to the exploration of the understandings that underlie interventions for children and youth.

Reviewer 2 Report

The purpose of this scoping review study was to: “…contribute to the social sustainability of LTPA programs through pointing out how communal understandings (that appear in line with understandings of equality or equity) can help guide stakeholders about the potentials to be pursued and barriers to be reduced in order to promote access to LTPA for children and youth.”  The study builds from a well cited background on disparities in leisure-time physical activity (LTPA) by socioeconomic status and education among children and youth, as well as a citation of gaps in exploring an equality and equity framework in relation to LTPA based previous review studies.  I found the paper to be innovative and to contribute an important theoretical perspective based on the equality and equity framework for understanding access and barriers for LTPA engagement among children and youth. Other key strengths include a generally well described methods section, well organized presentation of findings, insightful themes that emerged from the review, and overall strong writing and flow of the manuscript. 

  1. Study purpose: While this reviewer greatly appreciates the theoretical lens this study brings to understanding LTPA access via an equality and equity framework, I found the study objective statement a bit challenging to follow. For example, it is not clear what is meant by ‘communal understandings’. Does this refer to the ‘communal understandings’ based on the findings of previous studies? Secondly, I struggled with the use of the terms of ‘potentials’.  With regard to ‘potentials;, are the authors referring to ‘facilitating factors for access to and participation in LTPA among children and youth’?  I recommend more direct language in stating the overall study objective(s) of the review and use of more explicit language for specific constructs such as ‘potentials’ (e.g., “to examine/describe how an equality and equity framework has been operationalized/applied to interventions aimed at increasing access and reducing barriers to LTPA among children and youth”- or something to that effect). 
  2. Introduction: This reviewer appreciated the citation of literature on SES disparities in LTPA. Given that disparities in LTPA access are not limited to SES- as the authors find in their review, the authors may consider broadening their literature review in the background by citing other key disparities for LTPA access, such as those experienced by children and youth with physical or intellectual disabilities (i.e., add a sentence or two to acknowledge other disparities in LTPA beyond SES, recognizing the need to stay within word limits).
  3. Equality vs. Equity framework.
  • While the authors do a generally nice job with defining these terms, they may consider connecting further this framework (in the introduction or discussion) with a common framework within the field of public health of ‘universal’ vs. ‘targeted’ public health strategies, which has existed for many years. See, for example, a recent paper on this topic by Dodge (2020) in J Child Psychol Psychiatry.
  • Social Determinants of Health: In exploring the equity frame, did the authors consider applying further a social determinants of health framework to guide their review? (e.g., as relates to addressing transportation, income, internet access, time poverty, etc).  Might this framework be helpful to cite in the discussion for future directions on how ‘equity’ may be further addressed in planning for interventions aimed at increasing access and reducing barriers for LTPA? 
  1. Methods:

a.) Small suggestion: In the first sentence of the methods section, consider being more direct in stating the type of literature review “a scoping review” vs. just a ‘literature review’.  While the authors appropriately cite the important work by Arksey and O’Malley, I also recommend providing a brief definition of what a scoping review consists of. 

b.) Timeframe for literature search: I do not see information on the timeframe (over what years) the review was conducted. Can the authors clarify?

c.) Inclusion/exclusion criteria:  While the authors focus the review on children and youth, they refer on p.11, lines 374-377 to a study of women (citation 47) and young adults with disabilities (citation 48).  While this reviewer appreciates the overlap between some definitions of youth as per the United Nations as people between the ages of 15 and 24 and the definition of young adults, as the review is focused on children and youth, including samples of adults seems outside the scope.  I feel this issue can be addressed by further clarification of the age range of ‘youth’ in the methods, and acknowledgment that this age range includes ‘young adults’. 

d.) Themes that emerged from the scoping review.  This reviewer greatly appreciated the insightful themes that emerged in the scoping review.  This strength notwithstanding, it was not clear for me in the methods how authors identified key themes cited in the findings (e.g., “recruiting over change”, “reproducing social inequalities”). Can the authors provide a brief description in the methods regarding how these themes were identified? For example, was there a coding scheme used? Thematic or content analysis approach? Were studies each reviewed and coded separately by the two authors, and then themes developed?

  1. Findings: Again, this reviewer appreciated the good organization of the studies along with the rich themes that emerged.  These strengths notwithstanding, I found some themes were not immediately apparent in their write-up (e.g., “recruiting over change”, “local I”, “local II”).  Can the authors provide a little more detail on the meaning of these construct labels in the write-up?
  2. Other strengths: This reviewer greatly appreciated the insightful findings regarding the need for stakeholder perspectives and involvement in increasing equity and access for LTPA.

Author Response

Dear reviewer,

Thank you for giving us the opportunity to improve our manuscript. We very much appreciate the time and effort that you have dedicated to providing valuable feedback on the manuscript. Most of the suggestions you provided have led us to make changes. We have highlighted these changes within the manuscript. Here is a point-by-point response to your comments and concerns.

Comments from reviewer 2

Comment 1: Study purpose: While this reviewer greatly appreciates the theoretical lens this study brings to understanding LTPA access via an equality and equity framework, I found the study objective statement a bit challenging to follow. For example, it is not clear what is meant by ‘communal understandings’. Does this refer to the ‘communal understandings’ based on the findings of previous studies? Secondly, I struggled with the use of the terms of ‘potentials’.  With regard to ‘potentials;, are the authors referring to ‘facilitating factors for access to and participation in LTPA among children and youth’?  I recommend more direct language in stating the overall study objective(s) of the review and use of more explicit language for specific constructs such as ‘potentials’ (e.g., “to examine/describe how an equality and equity framework has been operationalized/applied to interventions aimed at increasing access and reducing barriers to LTPA among children and youth”- or something to that effect). 

Response: Thank you for pointing this out. We agree with this comment. Therefore, we have revised the section describing the objective of the study in order to clarify the meaning of central concepts accordingly. A clarification on communal understandings is also added in the introduction (page 2, lines 55-63). Furthermore, we have decided to use the term facilitating factors instead of potentials, and have incorporated this change throughout the manuscript.

Comment 2: Introduction: This reviewer appreciated the citation of literature on SES disparities in LTPA. Given that disparities in LTPA access are not limited to SES- as the authors find in their review, the authors may consider broadening their literature review in the background by citing other key disparities for LTPA access, such as those experienced by children and youth with physical or intellectual disabilities (i.e., add a sentence or two to acknowledge other disparities in LTPA beyond SES, recognizing the need to stay within word limits)

Response: We agree and have revised accordingly (page 2, line 45-47).

Comment 3a: While the authors do a generally nice job with defining these terms, they may consider connecting further this framework (in the introduction or discussion) with a common framework within the field of public health of ‘universal’ vs. ‘targeted’ public health strategies, which has existed for many years. See, for example, a recent paper on this topic by Dodge (2020) in J Child Psychol Psychiatry

Response: Thank you for this suggestion, which we think contributes to our definitions of the concepts of equity and equality. We have incorporated your suggestion in section 1.2 (page 3, lines 126-145).

Comment 3b: Social Determinants of Health: In exploring the equity frame, did the authors consider applying further a social determinants of health framework to guide their review? (e.g., as relates to addressing transportation, income, internet access, time poverty, etc).  Might this framework be helpful to cite in the discussion for future directions on how ‘equity’ may be further addressed in planning for interventions aimed at increasing access and reducing barriers for LTPA? 

Response: Thank you for this suggestion. We agree that the Social Determinants of Health framework is something that can add a valuable perspective to our use of the concepts of equity and equality, and have cited the framework in the discussion as you suggested (page 17, lines 525-527).

Comment 4a: Small suggestion: In the first sentence of the methods section, consider being more direct in stating the type of literature review “a scoping review” vs. just a ‘literature review’.  While the authors appropriately cite the important work by Arksey and O’Malley, I also recommend providing a brief definition of what a scoping review consists of. 

Response: We agree, and have incorporated your suggestion in the method section (page 4, lines 147-150).

Comment 4b: Timeframe for literature search: I do not see information on the timeframe (over what years) the review was conducted. Can the authors clarify?

Response: The dates of the literature search is stated in section 1.2 (page 4, lines 181-182). The timeframe used as inclusion/exclusion criteria is the years 2000-2020 stated in section 1.2 (page 4, lines 164) and section 2.2 (page 6, line 200).

Comment 4c: Inclusion/exclusion criteria:  While the authors focus the review on children and youth, they refer on p.11, lines 374-377 to a study of women (citation 47) and young adults with disabilities (citation 48).  While this reviewer appreciates the overlap between some definitions of youth as per the United Nations as people between the ages of 15 and 24 and the definition of young adults, as the review is focused on children and youth, including samples of adults seems outside the scope.  I feel this issue can be addressed by further clarification of the age range of ‘youth’ in the methods, and acknowledgment that this age range includes ‘young adults’

Response: We agree that this criterion needed clarification and have revised it accordingly in section 2.2. (page 6, lines 2003-2006).

Comment 4d: Themes that emerged from the scoping review.  This reviewer greatly appreciated the insightful themes that emerged in the scoping review.  This strength notwithstanding, it was not clear for me in the methods how authors identified key themes cited in the findings (e.g., “recruiting over change”, “reproducing social inequalities”). Can the authors provide a brief description in the methods regarding how these themes were identified? For example, was there a coding scheme used? Thematic or content analysis approach? Were studies each reviewed and coded separately by the two authors, and then themes developed?

Response: Thank you for pointing this out. We agree, and have incorporated a brief description of our process of identifying key themes in section 2.2. (page 6, lines 2019-230).

Comment 5: Findings: Again, this reviewer appreciated the good organization of the studies along with the rich themes that emerged.  These strengths notwithstanding, I found some themes were not immediately apparent in their write-up (e.g., “recruiting over change”, “local I”, “local II”).  Can the authors provide a little more detail on the meaning of these construct labels in the write-up?

Response: Thank your for pointing this out. We agree with this comment. In order to clarify the meaning of the themes “recruiting over change”, “local I”, “local II we have revised the headlines and provided more detail in the write ups of these themes in sections 3.1.2 (page 13, lines 320-335), 3.2.1 (page 14, lines 369-396).

Round 2

Reviewer 1 Report

You responded well to my comments. Thank you. No have no further issues.